# Synthesis, Characterization, Conformation in Solution, and Thermoresponsiveness of Polymer Brushes of methoxy[oligo (propylene glycol)-block-oligo(ethylene glycol)]methacrylate and N-[3-(dimethylamino)propyl]methacrylamide Obtained via RAFT Polymerization

**DOI:** 10.3390/polym15071641

**Published:** 2023-03-25

**Authors:** Maria Simonova, Denis Kamorin, Alexander Filippov, Oleg Kazantsev

**Affiliations:** 1Institute of Macromolecular Compounds of the Russian Academy of Sciences, Bolshoy Prospekt 31, 199004 Saint Petersburg, Russia; 2Research Laboratory “New Polymeric Materials”, Nizhny Novgorod State Technical University n.a. R.E. Alekseev, 24 Minin Street, 603950 Nizhny Novgorod, Russia

**Keywords:** synthesis, RAFT polymerization, dilute solution, static and dynamic light scattering, polymer brushes, macromolecular conformation, critical micelle concentration

## Abstract

The thermo- and pH-responsive polymer brushes based on methoxy[oligo(propyleneglycol)_8_-block-oligo(ethyleneglycol)_8_]methacrylate with different concentrations of N-[3-(dimethylamino)propyl]methacrylamide (from 0% to 20%) were synthesized via RAFT polymerization. The “grafting-through” approach was used to prepare the low-molar-mass dispersion samples (*M*_w_/*M*_n_ ≈ 1.3). Molar masses and hydrodynamic characteristics were obtained using static and dynamic light scattering and viscometry. The solvents used were acetonitrile, DMFA, and water. The molar masses of the prepared samples ranged from 40,000 to 60,000 g·mol^–1^. The macromolecules of these polymer brushes were modeled using a prolate revolution ellipsoid or a cylinder with spherical ends. In water, micelle-like aggregates were formed. Critical micelle concentrations decreased with the content of N-[3-(dimethylamino)propyl]methacrylamide. Molecular brushes demonstrated thermo- and pH-responsiveness in water–salt solutions. It was shown that at a given molecular mass and at close pH values, the increase in the number of N-[3-(dimethylamino)propyl]methacrylamide units led to an increase in phase separation temperatures.

## 1. Introduction

In recent decades, for the controlled delivery of various medicinal substances to diseased organs, various polymeric forms such as hydrogel nanocapsules, micelles, dendrimers, etc. have been proposed and intensively studied [1,2,3,4,5,6,7,8,9,10,11,12,13,14,15]. One of the most promising means of delivery is the micelles of polymers, in the hydrophobic core of which the drugs poorly soluble in water are retained, which are then released due to diffusion or destruction of micelles under external influence.

One convenient way to create a shell for polymer nanoparticles is using thermo- and pH-responsive polymers. One of them is polyethylene glycol (PEG) [16,17]. It allows the particle to circulate through the circulatory system for a long time and penetrate through various membranes and the blood–brain barrier [18]. The most rapidly developing method of obtaining polymer particles with a surface from polyethylene glycol fragments is the use of macromonomers, namely the derivatives of (meth)acrylic acid with ethylene glycol groups in the substituent [19,20]. Amphiphilic (co)polymers methoxiolygoethylenglycolmethacrylate (MOEGM) have good biocompatibility and low toxicity, are susceptible to biodegradation, and have low critical solution temperature, which is close to human body temperature. In [21], a series of novel temperature-responsive copolymer brushes with P-(2-(2-methoxyethoxy)ethyl methacrylate)-*co*-acrylamide) (P-(OEGMA188-*co*-AAm)) chains grafted from glass surfaces functionalized with (3-aminopropyl)triethoxysilane followed by the ATRP initiator were synthesized. P(OEGMA188-*co*-AAm) with a high mole fraction of AAm demonstrates “schizophrenic” behavior in wettability after immersion in pH buffer solutions, with transitions that mimic LCST and UCST for pH = 3, LCST for pH = 5 and 7, and temperature-induced transitions blocked for pH = 9.

For the synthesis of MOEGM copolymers with a given architecture such as block copolymers, brushes, star polymers, and dendrimers, as well as low-molar-mass distribution, the method of reversible addition–fragmentation chain transfer (RAFT) radical polymerization is used [22,23,24,25]. The copolymer poly(MEO_2_MA-co-PEGMA2080 (*M*_n_ = 17,300 g·mol^–1^) containing 2-(2-methoxyethoxy)ethylmethacrylate (MEO_2_MA, *M* = 188 g·mol^–1^) and poly(ethylene glycol) methyl ether methacrylate (PEGMA, *M* = 2080 g·mol^–1^) was synthesized using the atom transfer radical polymerization (ATRP) process, and its thermoresponsive behavior in water solution were studied. In comparison to other thermoresponsive random copolymers based on oligo(ethylene glycol) methacrylates, this copolymer exhibited an unusual thermally induced two-stage aggregation process. The copolymer chains associate during the first thermal transition and then undergo a rearrangement process at the second thermal transition to produce a stable core–shell micellar structure [19]. It was shown that PEG-b-(PAA-g-PLA) poly(ethylene glycol)-b-(polyacrylic acid)-g-poly(lactic acid), amphiphilic brush polymers synthesized via RAFT polymerization aggregated into spheres and vesicles with sizes of 70–110 nm in aqueous media [26]. Huang et al. synthesized asymmetric brush polymers (PtBA-g-PS)-co-PPEGMEA polytert-butyl2-((2-bromopropanoyloxy)-methyl)acrylate-g-polystyrene)-co-poly (ethylene glycol) methyl ether methacrylate via the RAFT polymerization of polyethylenglycole (PEG) methacrylate with tert-butyl 2-((2-bromopropanoyloxy)methyl) acrylate and ATRP of styrene. (PtBA-g-PS)-co-PPEGMEA formed large compound micelles in aqueous media, whereas (PAA-g-PS)-co-PPEGMA (polyacrylic acid)-g-polystyren)-co-poly(ethylene glycol) methacrylate self-assembled into bowl-shaped micelles with a hole at the edge of the micelles [27]. 

The effect of temperature and buffer solutions with different pH on the behavior of poly(oligo(ethylene glycol) methacrylate) (POEGMA) brush coatings, synthesized without the incorporation of the functional groups, was studied for the first time in detail using water contact angle measurements and atomic force microscopy. The thermoresponsiveness of the grafted brush coatings based on POEGMAs is driven by the LCST phenomenon. The obtained AFM results suggest a strong impact of the buffer solutions on the values of LCST transition and contact angle ranges, as well as on the morphology of the coatings. The ellipsometry data reflect the penetration of salt ions from buffer solutions into the brush coatings. In contrast to the “typical” behavior of POEGMA coatings in water, the different mechanisms available below LCST in the buffer solutions destroy the hydrated layers surrounding POEGMA macromolecules, leading to their collapse [28].

To summarize, it should be noted that research is actively underway on the development of new types of biocompatible polymeric stimulus-sensitive molecular brushes with controlled conformational and phase transitions in aqueous solutions, whose micelles can be used as nanocontainers for the delivery of hydrophobic drugs. In this regard, new water-soluble macromonomers, such as the esters of methacrylic acid with a diblock alcohol moiety containing hydrophilic oligoethylene glycol and hydrophobic oligopropylene glycol blocks, were used to synthesize polymer brushes. A distinctive feature of such macromonomers is the possibility of the fine regulation of their amphiphilic nature by varying the length and arrangement of hydrophilic and hydrophobic blocks, which should be expressed in the manifestation of the amphiphilic properties of polymers.

In previous studies [29,30], we have investigated polymethacrylic molecular brushes with oligo(ethylene glycol)-block-oligo(propylene glycol) side chains, which were obtained via conventional radical polymerization. In thermodynamically good solvents, namely acetonitrile, the investigated copolymers had a high intramolecular density, and the shape of their molecules resembled a star-shaped macromolecule. Phase separation temperatures were reduced with an increase in the content of the oligo(propylene glycol) block. 

To assess pH responsitivity, N-[3-(dimethylamino) propyl]methacrylamide (DMAPMA) was introduced into the polymer chain. DMAPMA is very hydrophilic and does not show thermosensitivity; therefore, the DMAPMA monomer increases the phase separation temperatures of copolymer solutions. The aim of the present work is to investigate the effect of the content of N-[3-(dimethylamino)propyl]methacrylamide on the hydrodynamic and conformational characteristics of thermoresponsive methoxy [oligo(propyleneglycol)_8_-block-oligo(ethyleneglycol)_8_] methacrylate and N-[3-(dimethylamino)propyl]methacrylamide (polyOPG_8_OEG_8_MA-DMAPMA) in dilute solutions. The structural formulae of homo- and copolymers are presented in Figure 1. 

## 2. Experimental 

### 2.1. Materials and Methods of Synthesis

The “grafting-through” method was used to produce polymers with a brush structure.

The synthesis of the polymer brushes using this approach involves a one-step process using macromonomers capable of radical polymerization. The macromonomer methoxy[oligo(propylene glycol)-block-oligo(ethylene glycol)] methacrylate with average lengths of oligo(propylene glycol) (*p*) and oligo(ethylene glycol) (*e)* fragments equal *p* = 7.9 and *e* = 8.2 was used to obtain the polymers. 

For the synthesis of the macromonomers, a previously described method involving involves the esterification of methacrylic acid with methoxy oligo(alkylene glycol)s [31,32] was used. The synthesis was carried out at a temperature of 120–125 °C in 30 wt% toluene solution in the presence of 2 wt% of p-toluene sulfonic acid as a catalyst and 0.3 wt% of hydroquinone as a polymerization inhibitor. Previous to polymerization, the macromonomer was passed through a basic alumina column to remove inhibitors.

The RAFT copolymerization of OPG_8_OEG_8_MA and DMAPMA was carried out using 4-cyano-4-((dodecylsulfanylthiocarbonyl)sulfanyl)pentanoic acid (CTA) as the RAFT agent and 2,2′-azobisisobutyronitrile (AIBN) as the initiator under the conditions of [Monomers]_0_:[ CTA]_0_:[AIBN]_0_ = 200:4:1 molar ratio in toluene at 60 °C. The total initial monomer concentration was 30 wt%. The initial ratio of macromonomer to amine-containing monomer varied from 100:0 to 80:20. The structures of the copolymers OPG_8_OEG_8_MA-DMAPMA were confirmed via nuclear magnetic resonance (NMR) spectroscopy (DDR2 400; Agilent, Santa Clara, CA, USA) in CHCL_3_-D6.

### 2.2. The Determination of CMC

The critical micelle concentrations (*CMCs*) of copolymers were determined via fluorimetry using pyrene as a fluorescent probe [33,34]. The steady-state fluorescence spectra were recorded on a Shimadzu RF-6000 spectrofluorimeter (Shimadzu, Kyoto, Japan) at the temperature of 25 °C.

### 2.3. Methods of Molecular Hydrodynamics and Optics

The absolute molar masses (MMs) and the hydrodynamic radii *R*_h-D_ of macromolecules were determined via static (SLS) and dynamic light scattering (DLS) methods in dilute solutions in acetonitrile (density ρ_0_ = 0.78 g∙cm^−3^, dynamic viscosity η_0_ = 0.34 cP, and refractive index *n*_0_ = 1.341), DMFA (ρ_0_ = 0.94 g∙cm^−3^, η_0_ = 0.80 cP, and *n*_0_ = 1.428) and water (ρ_0_ = 1.00 g∙cm^−3^, η_0_ = 0.89 cP, and *n*_0_ = 1.333). The experiments were performed using a Photocor Complex instrument (Photocor Instruments Inc., Moscow, Russia), which is equipped with a Photocor DL diode laser (wavelength *λ* = 632.8 nm and power 5–30 mW). The instrument was calibrated using benzene (*R*_V_ = 2.32·10^−5^ cm^–1^). The measurements were carried out at scattering angles ranging from 45° to 135°. The solutions, solvents, and calibration were filtered into cells that were dust-free previously by benzene. Millipore filters (Millipore Corporation, Bedford, MA 01730 USA) with a PTFE membrane with a pore size of 0.45 μm were used. 

For the solutions investigated in acetonitrile and DMFA, the distribution of the light scattering intensity *I* over the hydrodynamic radii *R*_h-D_(*c*) of scattering objects was unimodal. The values of *R*_h-D_(*c*) were determined in the wide concentration range and extrapolated to zero concentration to obtain the hydrodynamic radius *R*_h-D_ of macromolecules. As is well known, the translation diffusion coefficients *D*_0_ and the friction coefficient *f* of macromolecules are related to *R*_h-D_, which is defined using Stokes–Einstein equations [35,36,37]:*D*_0_ *= k*_B_*T/f = k*_B_*T/6*πη_0_*R*_h-D_(1)
where *k*_B_ is Boltzmann’s constant and *T* is the absolute temperature.

SLS measurements were performed at the angle of 90° since no angular dependence of the scattered light was observed. The obtained results were analyzed according to the Debye method, and the values of the weight-average molar masses *M*_w_ and the second virial coefficient *A*_2_ were calculated using the following formula:(2)$$\frac{cH}{I_{90}}=\frac{1}{M_{w}}+2A_{2}c$$
where *H* is the optical constant.
(3)$$Н=\frac{4\pi^{2}n_{{}_{0}}^{2}{\left(dn/dc\right)}^{2}}{N_{A}\lambda_{0}^{4}}$$

Here, *I*_90_ is the excessive intensity of light scattered at an angle of 90°, *N*_A_ is Avogadro’s number, and *dn*/*dc* is the refractive index increment. The values of *dn*/*dc* were determined using an RA-620 refractometer (Shimadzu, Kyoto, Japan) with a wavelength *λ*_0_ = 589.3 nm. The values of *dn*/*dc* were calculated from the slope of concentration dependence on the difference between the refractive indexes of the solution *n* and the solvent *n*_0_ (Δ*n* = *n − n*_0_).

The viscometry experiments were performed using an Ostwald-type Cannon–Manning capillary viscometer (Cannon Instrument Company Inc., State College, PA, USA). The dependencies of the reduced viscosity η_sp_/*c* on the concentration were analyzed using the Huggins equation:η_sp_/*c* = [η] + *k*_H_[η]^2^*c*(4)
where [η] is the intrinsic viscosity, and *k*_H_ is the Huggins constant.

Light scattering, viscometry, and refractometry experiments were carried out at 21 °C. Millipore filters (Millipore Corp., Billerica, MA, USA) with a PTFE membrane with a pore size of 0.20 nm were used.

Moreover, the molar mass characteristics of polyOPG_8_OEG_8_MA and polyOPG_8_OEG_8_MA-DMAPMA were obtained via gel permeation chromatography (GPC), using gel permeation chromatography (Chromos LC-301, Chromos Engineering Co. Ltd., Dzerzhinsk, Russia) with an instrument isocratic pump, a refractometric detector, and two exclusive columns: Phenogel 5u 50A and 10^3^A (Phenomenex, Torrance, CA, USA) in tetrahydrofuran (THF). 

### 2.4. Investigation of Self-Assembly of polyOPG_8_OEG_8_MA-DMAPMA in Aqueous Solutions

The aqueous solutions of the copolymers were investigated using the methods of light scattering and turbidimetry with the Photocor Complex described above, which is also equipped with a Photocor-PD detection device for measuring the transmitted light intensity. The solution temperature *T* was changed discretely, with the step ranging from 1.0 to 5.0 °C. At steady-state conditions, i.e., when the solution parameters do not depend on time, the hydrodynamic radii *R*_h_ of scattering species and their contribution *S*_i_ to the integral scattering intensity were determined. *S*_i_ was estimated using the values of the areas under the curve of the corresponding *R*_h_ distribution peak. For all copolymers, the polymer concentration was *c* = 0.0050 g∙cm^−3^, and polyOPG_8_OEG_8_MA-DMAPMA 90:10 was investigated in the concentration range from *c* = 0.0025 to 0.0100 g∙cm^−3^. 

A phase transition temperature (*T*_ph_) was determined from the temperature dependence of optical transmittance.

The acidity of the pH medium varied from 3.6 to 12.4 in buffer solutions (pH 3.6, 6.86, 12.4, Hanna Instruments) and the pH of the water solution was determined using a pH meter (Sartorius, Finland) and pH-meter-ionomer Expert-001 (Russia).

## 3. Results and Discussion

### 3.1. Synthesis, Structure, Molar Masses, and Hydrodynamic Characteristics of polyOPG_8_OEG_8_MA-DMAPMA 

The samples of the copolymers were synthesized using RAFT polymerization. Their structure was confirmed using NMR spectroscopy and GPC (Figure 2 and Figure 3).

The values of the refractive index increments increased with an increase in the number of DMAPMA units. Moreover, in both acetonitrile and DMFA, the *dn/dc* dependencies on DMAPMA fractions were well illustrated in a straight line. Therefore, for the studied samples, the principle of the additivity of the refractive index increments of monomeric units was considered.

According to the chromatography method, the polydispersity indexes *Đ* = *M*_w_/*M*_n_ of the prepared samples were close (Table 1). On the other hand, the values of molar masses obtained using GPC and SLS did not coincide. This difference is likely due to the fact that the GPC method does not allow one to obtain correct values for polymers with complex architecture, in particular molecular brushes. 

The molar masses of the investigated polymers were determined in acetonitrile. Unfortunately, it was not possible to measure the MM in DMF, due to the low value of the refractive index increment *dn*/*dc*, which ranged from 0.03 to 0.037 cm^3^·g^–1^. In acetonitrile, *dn*/*dc* changed from 0.120 to 0.137 cm^3^·g^–1^. It is worth noting that the *M*_w_ values of copolymer samples differed insignificantly.

Using the MM values, it is easy to calculate the polymerization degree *N*_b_ of the backbone of the copolymers according to the following equation:*N*_b_ = *M*_w_/*M*_0-cp_(5)
where *M*_0-cp_ values indicate the molar masses of the repeating units of polyOPG_8_OEG_8_MA and polyOPG_8_OEG_8_MA-DMAPMA. The values of *M*_0-cp_ (Table 2) are determined using copolymer composition (*x* and *y*, see Figure 1) and molar masses *M*_0-1_ = 904 and *M*_0-2_ = 170 g∙mol^−1^ of OPG_8_OEG_8_MA and DMAPMA, respectively: *M*_0-cp_ = (*x*∙*M*_0-1_ + *y*∙*M*_0-2_)/(*x*+*y*)(6)

The *N*_b_ values are listed in Table 2. Table 2 also presents the average values of the length *L*_b_ = *N*_b_∙λ_0-b_ of the backbone. Length *L*_b_ is calculated under the assumption that all valence bonds have the same length of 0.14 nm and that the valence angles are tetrahedral. Consequently, the length of the repeating unit of the main chain was λ_0-b_ = 0.25 nm. The *L*_b_ values were only 2–3 times greater than the length *L*_sc_ = 6.4 nm of the side chain of the OPG_8_OEG_8_MA monomer. The chains of the second component were much shorter, and their length was *L*_DMAPMA_ = 0.9 nm. The *L*_sc_ and *L*_DMAPMA_ values were calculated using the described assumptions. Notably, when *N*_b_ and *L*_b_ were estimated, the molar masses of the terminal groups of the main chain and their length were not taken into account. The MM of these groups was about 460 g∙mol^−1^, which was about 1 percent of the MM of the lowest molecular weight sample polyOPG_8_OEG_8_MA-DMAPMA 95:5. Accordingly, the actual value of ***N*_b_** of the studied samples differed from the values presented in Table 2 by less than one percent. The total length of the end groups was about 2.8. nm, i.e., two times less than *L*_sc_.

The obtained structural parameters allowed us to make preliminary assumptions about the shape of the copolymer macromolecules. Figure 3 shows a simplified molecular schema polyOPG_8_OEG_8_MA-DMAPMA 80:20. It is clearly seen that the transverse and longitudinal dimensions did not significantly differ. The “diameter” ***L*_⊥_** = 2***L*_sc_** of the macromolecule was determined by the length ***L*_sc_** of the OPG8OEG8MA side chains and did not exceed 13 nm. The largest longitudinal dimension ***L*** _‖_ was equal to the sum of the backbone length and twice the side chain length ***L*_b_** + 2***L*_sc_**. This conformation of the macromolecule was realized in the “ideal” case, when both the main and side chains were in a fully extended trans-conformation. Naturally, the real situation was somewhat different. Indeed, the side chains were quite long, and they were most likely more or less folded. For steric reasons, the side chains located near the terminal groups were probably folded much more strongly than the chains located in the central part of the backbone. Accordingly, Δ ≤ δ ˂ ***L*_sc_** (Figure 4).

In the first approximation, the macromolecules were modeled using a revolution prolate ellipsoid or a cylinder with spherical ends. The asymmetry parameter *p* or the ratio of longitudinal to transverse dimensions is determined by the following equation:*p = **L*** _‖/_***L***_⊥_ = (***L*_b_**+2**Δ**)/2 δ ˂ ***L*_b_**/2***L*_sc_** + Δ/δ(7)
where the ratio *Δ*/δ is less than unity. For the studied copolymers, the value of ***L*_b_**/2***L*_sc_** ranged from 0.9 for polyOPG_8_OEG_8_MA-DMAPMA 95:5 to 1.4 polyOPG_8_OEG_8_MA-DMAPMA 80:20. Accordingly, for all the studied samples, *p* < 2.4.

For the studied polymers, low experimental values of both the intrinsic viscosity and hydrodynamic radii of macromolecules were obtained. In principle, this could be expected since the macromolecules of the copolymers under consideration had a dense structure. On the other hand, the calculations using the formulas for a rigid ellipsoid of revolution and a cylinder (see monograph [35] and the references in it) and the obtained structural parameters ***L*_b_** and ***L*_sc_** resulted in the values of [η] and ***R*_h-D_**, which strongly differed from the experimental values of these characteristics. Therefore, for the studied copolymers, the use of the rigid particle models for the analysis of hydrodynamic characteristics is incorrect, although these models quite often adequately describe the hydrodynamic behavior of polymers with complex architecture in dilute solutions [35]. It can be assumed that, in this case, an important role is played not only by the permeability of macromolecules but also by the change in their shape due to the difference in the real conformation of the side chains from the conformation of the trans-chain. Analyzing Figure 3, one would expect that an increase in the DMAPMA units should not lead to a significant decrease in intramolecular density. However, the observed changes in these characteristics were sufficient to change the hydrodynamic invariant *A***_0_**, which was calculated using the formula in [35,38,39]. This characteristic is determined with the experimental values of molar mass *M*, intrinsic viscosity [η], and translation diffusion coefficient *D*_0_ as follows:
(8)$$A_{0}=\eta_{0}D_{0}{\left(\frac{M\left[\eta \right]}{100}\right)}^{1/3}/T$$
where *η*_0_ is the viscosity of the solvent, and *T* is the absolute temperature. 

As can be seen from Table 1, *A_0_* values increased from 2.4 to 3.2 **×** 10^10^, erg·K^−1^mol^−1/3^ with increasing DMAPMA content, i.e., a decrease in intramolecular density. Similar behavior was previously observed in the homologous series of star-shaped four-armed poly-2-ethyl-2-oxazine [40]. Note that low values of *A_0_* ≤ 2.8, i.e., lower than the theoretically predicted value for a hard sphere, are typical for polymers with complex architecture, such as molecular brushes, as well as hyperbranched and star-shaped polymers [41,42,43,44]. 

### 3.2. Characteristics of polyOPG_8_OEG_8_MA-DMAPMA in Aqueous Solutions at Room Temperatures

At 21 °C, two modes were detected using DLS for the aqueous solutions of the investigated polymer brushes. For all samples at all concentrations, on average, the hydrodynamic radii *R_h-f_* (Table 3) of the particles responsible for fast mode exceeded the hydrodynamic radius *R_h-D_* of macromolecules determined in acetonitrile, by 30 percent (Table 2). Therefore, the species with radius *R_h-f_* had supramolecular structures. This fact can be explained by the formation of micelles in water. The *CMC* values for polymers in aqueous solutions are presented in Table 3. Note that an increase in the DMAPMA fraction led to an increase in the *CMC*, i.e., the introduction of more hydrophilic DMAPMA units reduced the tendency of the polymer to aggregate due to changing the hydrophilic–hydrophobic balance of the molecules. However, the change in *CMC* values was not significant. 

The objects responsible for the slow mode were aggregates with hydrodynamic radius *R_h-s_*. The hydrodynamic radii of supramolecular structures *R_h-s_* were more than an order of magnitude greater than the size of the isolated macromolecules *R_h-D_* and micelles *R_h-f_*. This fact indicates that a very large number of polymer molecules were combined into aggregates. Note that for homo polyOPG_6.6_OEG_8.3_MA obtained using radical polymerization, the aqueous solutions, or more precisely, only the micelles in them, were unimodal [30]. Hence, it can be assumed that the terminal groups of the copolymers obtained via RAFT polymerization play a significant role in the formation of large aggregates. The relative weight concentration (*c*_s_) of large aggregates was much less than the concentration (*c*_f_) of micelles. Indeed, the estimate in terms of hard-sphere models for micelles and coil for large aggregates revealed that *c*_s_ was less than 10 percent (Table 3).

### 3.3. Characteristics of Aqueous Solutions of polyOPG_8_OEG_8_MA-DMAPMA on Heating

Figure 5 shows the temperature dependencies of the relative light scattering intensity *I*/*I*_21_ and transmitted intensity *I**/*I**_21_, the hydrodynamic radii, and ratio *S*_s_*/S*_f_ for the aqueous solution polyOPG_8_OEG_8_MA-DMAPMA 90:10. (*I*_21_ and *I**_21_ are light scattering intensity and optical transmission at 21 °C, and *S*_s_ and *S*_f_ are contributions of large aggregates and micelles to the integral light scattering intensity of the solution).

Similar dependencies were obtained for all the studied samples at all concentrations. Several temperature intervals can be distinguished on these dependencies. The first of them occurred at *T* ≤ *T*_1_, when the optical transmission did not depend on the temperature. At *T*_1,_ a sharp decline in *I** was observed. Accordingly, *T*_1_ marked the onset of the phase separation interval. In the third temperature interval at *T* ≤ *T*_2_, *I** = 0, and *T*_2_ was considered the temperature of the finishing phase transition.

The temperatures of phase separation were determined by analyzing the dependencies of *I* on *T*. At the temperature of the phase separation onset, a sharp increase in the light scattering intensity was observed. *I* reached the maximum value at the temperature of the finishing phase transition. Further heating was accompanied by a decrease in the *I* value. The phase transition temperatures determined via turbidimetry and SLS coincided with an accuracy of one degree.

In the first temperature interval, the *I* values decreased, which was caused by a slight change in the hydrodynamic radii *R_h-s_* of the aggregates (Figure 5). This is probably due to the partial dehydration of the side chains and the formation of intramolecular bonds with increasing temperature. It can be assumed that a similar process also occurred in micelles. However, their small size did not allow one to record the changes using DLS, and at *T* ≤ *T*_1_, no change was observed in the hydrodynamic radius *R_h-f_*. Correspondingly, the contribution of the aggregates to the integral intensity of the scattered light decreased (Figure 3). 

At *T*_1_ ≤ *T* ≤ *T*_2_, the size of the aggregates strongly increased, while micelles were no longer visible using DLS. Therefore, in the phase separation interval, aggregation occurred, and hydrodynamic radii exceeded one micron. At *T* ≥ *T*_2_, the light scattering intensity and hydrodynamic radii of the aggregates decreased, which was caused by the precipitation of the part of the polymer in the sediment.

### 3.4. The Dependence of Phase Separation Temperatures on the Concentration of polyOPG_8_OEG_8_MA-DMAPMA 90:10 Solution

Table 4 presents the phase separation temperatures for a solution of polyOPG_8_OEG_8_MA 90:10 with a concentration ranging from 0.0025 g∙cm^−3^ to 0.01 g∙cm^−3^. It was found that with an increase in concentration, the phase separation temperatures decreased, i.d. with dilution, and the quality of the solvent improved the size of formed aggregates at room temperature and at temperature *T*_1_ decreased; that is, the limits of solubility increased. We did not observe any change in macromolecule radii, since they were small. The aggregates decreased with dilution. Similar behavior has been reported for other thermo- and pH-sensitive polymers [23].

### 3.5. The Influence of Composition of Copolymers on Phase Separation Temperatures at Fixed Concentration and pH Solutions

The influence of the composition in a wide range of water solutions at *c* = 0.005 g∙cm^−3^ on the properties of solutions of polyOPG_8_OEG_8_MA-DMAPMA was observed at room temperature. The size of fast and slow modes with a decrease in the DMAPMA content grew.

The molar masses of the investigated copolymers, pH, and concentration (0.005 g∙cm^−3^) of solutions changed insignificantly in the series of copolymers; it became possible to compare the phase separation temperatures.

It was found that with the increase in the number of DMAPMA units, the temperatures significantly increased at pH 3.56, slightly increased at pH 6.86, and did not change at pH = 12.43.

Note that polyOPG_8_OEG_8_MA is not pH-sensitive, but the determined phase separation temperatures at different pH values did not coincide. This fact is related to the effect of salting out, which occurs to a noticeable extent at low contents of ionogenic monomer. In the case of a greater number of DMAPMA units, as expected, the phase separation temperatures decreased with increasing pH (Table 5). 

## 4. Conclusions

The thermo- and pH-responsive polymer brushes based on methoxy[oligo(propyleneglycol)_8_-block-oligo(ethyleneglycol)_8_]methacrylate with different concentrations of N-[3-(dimethylamino)propyl]methacrylamide (from 0% to 20%) were successfully synthesized via RAFT polymerization. The “grafting-through” approach was used to prepare the low-molecular-weight dispersion samples (*M*_w_/*M*_n_ ≈ 1.3). Molar masses and hydrodynamic characteristics were obtained using static and dynamic light scattering and viscometry. The solvents used were acetonitrile, DMFA, and water. The solutions in acetonitrile were molecularly dispersed. The molar masses of the prepared samples ranged from 40,000 to 60,000 g·mol^–1^. It was established that for all copolymers, the side chains of N-[3-(dimethylamino)propyl] methacrylamide shield the backbone and decrease intermolecular density. Analyzing characteristics such as molar masses, hydrodynamic radius (diffusion coefficient), and intrinsic viscosity, we concluded that, in the first approximation, the macromolecules of polymer brushes based on methoxy[oligo(propyleneglycol)_8_-block-oligo(ethyleneglycol)_8_]methacrylate with different concentrations of N-[3-(dimethylamino)propyl]methacrylamide could be modeled using a prolate revolution ellipsoid or a cylinder with spherical ends. In water, micelle-like aggregates were formed. Critical micelle concentrations decreased with the content of N-[3-(dimethylamino)propyl]methacrylamide. Molecular brushes demonstrated thermo- and pH-responsiveness in water–salt solutions. Our findings reveal that at a given molecular masse and at close pH values, an increase in the number of N-[3-(dimethylamino)propyl]methacrylamide units leads to an increase in the phase separation temperatures. 

## Figures and Tables

**Figure 1 polymers-15-01641-f001:**
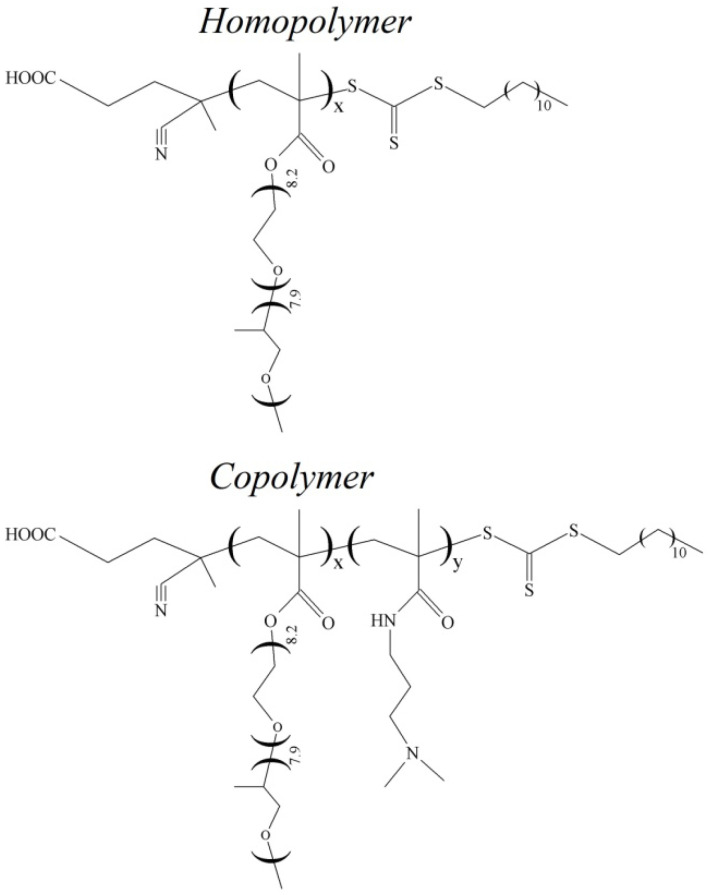
Structures polyOPG_8_OEG_8_MA and polyOPG_8_OEG_8_MA-DMAPMA.

**Figure 2 polymers-15-01641-f002:**
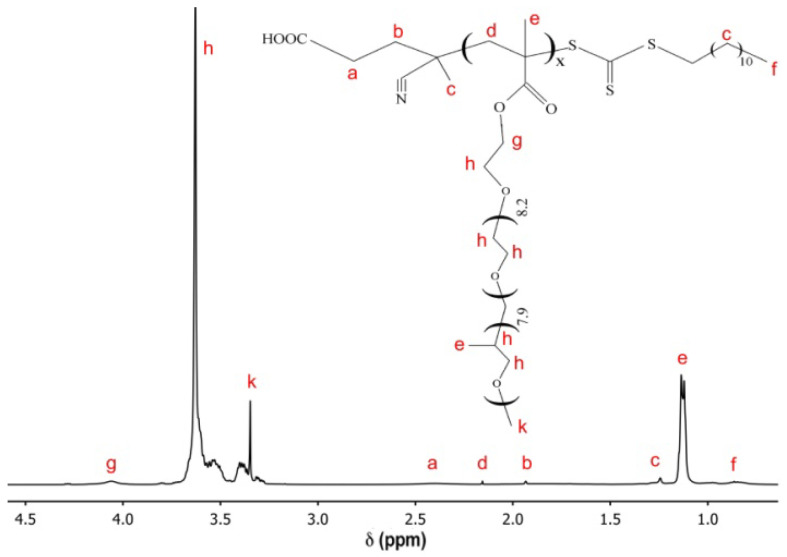
^1^H NMR spectra of polyOPG_8_OEG_8_MA.

**Figure 3 polymers-15-01641-f003:**
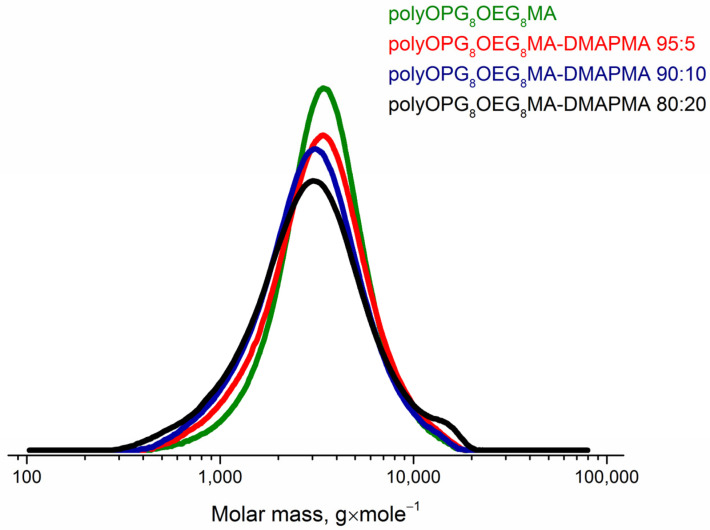
GPC traces of the polymers.

**Figure 4 polymers-15-01641-f004:**
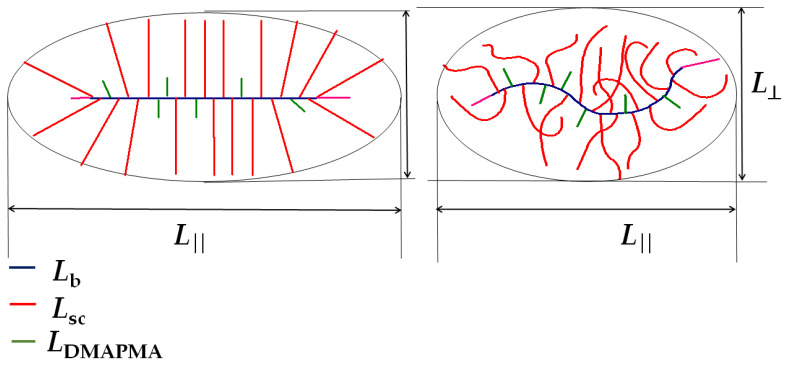
The simplified molecular image for polyOPG_8_OEG_8_MA-DMAPMA 80:20.

**Figure 5 polymers-15-01641-f005:**
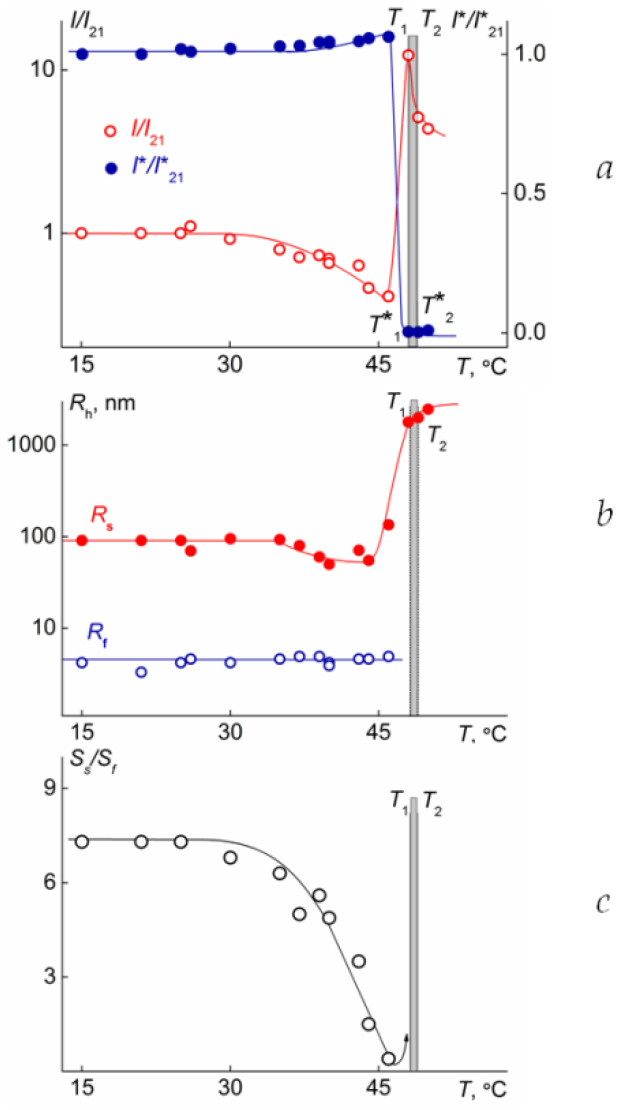
Temperature dependencies of the relative light scattering intensity *I*/*I*_21_ and transmitted intensity *I**/*I**_21_, (**a**) the hydrodynamic radii (**b**), and *S*_s_*/S*_f_ (**c**) for the water solution polyOPG_8_OEG_8_MA at concentration *c* = 0.01 g∙cm^−3^. *I*_21_ is light scattering intensity at 21 °C.

**Table 1 polymers-15-01641-t001:** Molar mass and hydrodynamic characteristics of polymer brushes.

Solvents	*M*_w_ × 10^−3^, g·mol^−1^ SLS	*R*_h-D,_ nm	[η], cm^3^·g^−1^	A_0_ × 10^10^, erg·K^−1^mol^−1/3^	*M*_w_ × 10^−3^, g·mol^−1^ SEC (in THF)	*Đ* (in THF)
polyOPG_8_OEG_8_MA-DMAPMA 80:20						
acetonitrile	55	4.2	8.7	3.20	16	1.4
DMFA	-	2.7	-			
water		5.8 *				
polyOPG_8_OEG_8_MA-DMAPMA 90:10						
acetonitrile	50	3.9	7.2	2.90	15	1.3
DMFA	-	2.2	-			
water		5.4 *				
polyOPG_8_OEG_8_MA-DMAPMA 95:5						
acetonitrile	40	3.9	8.8	2.85	16	1.3
DMFA	-	3.0				
water		5.2 *				
polyOPG_8_OEG_8_MA						
acetonitrile	50	4.2	5.4	2.40	17	1.2
DMFA		3.5				
water		4.9 *				

** R*_h m_ size of micelle-like aggregates of investigated copolymers.

**Table 2 polymers-15-01641-t002:** The structure characteristics of polyOPG_8_OEG_8_MA and polyOPG_8_OEG_8_MA-DMAPMA.

Sample	*M*_0-cp_, g∙mol^−1^	*M*_w_ × 10^−3^, g∙mol^−1^	*N* _b_	*L*_b_, nm	*L*_sc_, nm	*L*_DMAPMA,_ nm
polyOPG_8_OEG_8_MA-DMAPMA 80:20	757	55	73	18.4	6.4	0.9
polyOPG_8_OEG_8_MA-DMAPMA 90:10	831	50	60	15.1	6.4	0.9
polyOPG_8_OEG_8_MA-DMAPMA 95:5	867	40	46	11.6	6.4	0.9
polyOPG_8_OEG_8_MA	904	50	55	13.9	6.4	0.9

**Table 3 polymers-15-01641-t003:** Characteristics of solutions of polyOPG_8_OEG_8_MA-DMAPMA at *c* = 0.005 g∙cm^−3^ in water.

Samples	pH	*T*_1_, °C	*T*_2_, °C	*R_h-f_* Room, nm	*R_h-s_* Room, nm	Δ*T*, °C	*T* ph °C	*c* _f_	*CMC* *wt%*
polyOPG_8_OEG_8_MA-DMAPMA 80:20	7	54	67	4.0	80.3	13	50	90	0.0013
polyOPG_8_OEG_8_MA-DMAPMA 90:10	7	46	56	4.2	83	10	50	96	0.0011
polyOPG_8_OEG_8_MA-DMAPMA 95:5	6	45	49	4.9	50	4	48	96	0.00057
polyOPG_8_OEG_8_MA	6	44	47	5.4	-	3	46	96	0.00045

**Table 4 polymers-15-01641-t004:** Phase separation temperatures for polyOPG_8_OEG_8_MA 90:10.

Concentration, g∙cm^−3^	pH	*T*_1_, °C	*T*_2_, °C	*R_h-F_* Room	*R_h-s_* Room	Δ*T*, °C
polyOPG_8_OEG_8_MA 90:10						
0.25	7	44	56	5.4	70	12
0.5	7	42	53	4.2	86	11
1	7	40	50	3.3	98	10

**Table 5 polymers-15-01641-t005:** Phase separation temperatures for solution of polyOPG_8_OEG_8_MA-DMAPMA at different pH values.

Samples	pH = 3.56	pH = 6.86	pH = 12.43
polyOPG_8_OEG_8_MA-DMAPMA 80:20	56.0	49.0	46.8
polyOPG_8_OEG_8_MA-DMAPMA 90:10	50.1	46.5	46.8
polyOPG_8_OEG_8_MA-DMAPMA 95:5	47.4	45.1	47.0
polyOPG_8_OEG_8_MA	45.4	44.1	47.6

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
