# Peer review of "Synthesis, Characterization, Conformation in Solution, and Thermoresponsiveness of Polymer Brushes of methoxy[oligo (propylene glycol)-block-oligo(ethylene glycol)]methacrylate and N-[3-(dimethylamino)propyl]methacrylamide Obtained via RAFT Polymerization"

_polymers, 2023, doi:10.3390/polym15071641_

Round 1

Reviewer 1 Report

Simonova et al. submitted the manuscript "Synthesis, characterization, conformation in solution and thermoresponsiveness of polymer brushes of methoxy[oligo  (propylene glycol)-block-oligo(ethylene glycol)]methacrylate and N-[3-(dimethylamino)propyl]methacrylamide obtained by  RAFT polymerization" derived the low molecular weight dispersive samples, where the physical parameters were evaluated using viscometry, and light scattering experiments.

There are a few points that are not discussed in the paper, and the authors need to revise them.

1. Was there any rationality behind using only acetonitrile and DMFA solvents while not using other organic solvents?

2. Were there controls commercial available that can be directly compared with the current results using the same experiments?

3. Authors mentioned that samples prepared in their study were characterized by NMR and IR spectroscopy; however, no NMR and IR spectra or characterization was indicated in the manuscript.

Minor comments.

-There are some typographical errors in the manuscript; please maintain uniformity in the paper.

-The naming of these prepared samples needs to be rechecked again

Author Response

  1. Was there any rationality behind using only acetonitrile and DMFA solvents while not using other organic solvents?

In the study of homopolymer systems polyOPG6.6OEG8.3MA, the chloroform, THF, water, acetonitrile were used. It was shown that using of acetonitrile makes it possible to obtain molecularly dispersed solutions. In this solvent we determined molar mass and hydrodynamic radii. In chloroform and water the micelle formation was detected. This fact was confirmed by fluorescent method (CMC).  When comparing the sizes of micelles and macromolecules, it was found that micelles formed in chloroform consist of 2-3 macromolecules. Unfortunately, THF is not a very convenient solvent for light scattering studies because its characteristics change under light. So in present article which devoted to investigation copolymers polyOPG8OEG8MA-DMAPMA the same solvents were used to compare the results obtained for these systems.

The important task our study was to prove the additivity of copolymer blocks with different fraction of pH-sensitive component DMAPMA. That is, long-term experiments on refractometry are necessary. In DMFA, one mode was also fixed, but the value of the increment of the refractive index was low, however, in this solvent, we only confirmed the additivity of the increments of the refractive index of the components of the block copolymer.

  1. Were there controls commercial available that can be directly compared with the current results using the same experiments?

To control the work of equipment (Photocor Instruments Inc., Moscow, Russia), we use polystyrene standards of various molecular weights and, accordingly, with different sizes of scattering particles (macromolecules).

  1. Authors mentioned that samples prepared in their study were characterized by NMR and IR spectroscopy; however, no NMR and IR spectra or characterization was indicated in the manuscript.

We present the NMR spectra and GPC data in the main text of the article. Figures 2 and 3, page 7.

Minor comments.

-There are some typographical errors in the manuscript; please maintain uniformity in the paper.

-The naming of these prepared samples needs to be rechecked again

Thank you. We checked the article for typos and the name of the polymers.

All changes in the article are highlighted in yellow

Reviewer 2 Report

The findings are very intriguing, but the paper needs to be significantly improved before it can be published. The following issues should be clarified. 

Please suggest what types of buffers were used in the studies.

The factor of the buffer solutions plays an important role in the pH- responsivity as well as the thermo-responsivety of the systems presented here. In a similar study [https://doi.org/10.1007/s00396-022-04959-1], it was shown that, in contrast to the "typical" behavior of POEGMA coatings in water, different mechanisms available in buffer solutions destroy hydrated layers surrounding POEGMA macromolecules, leading to their collapse.

In contrast to the first section of the article, which does a great job of explaining the synthesis and structure of the polyOPG8OEG8MA-DMAPMA, questions about the molecule's behavior at various concentrations, temperatures, and pH levels keep coming up. I wholeheartedly concur that this system is extremely complex, but efforts should be made to explain the molecular mechanisms of temperature-induced phase separation (types of intramolecular and intermolecular bonds).

Please provide tables 2, 3, and 4 with additional information. What do T1 and T2, ΔT and T ph, and other parameters mean?

Unfortunately, in a lot of places, the authors state only phenomena without trying to describe their nature. Which determines the thermoresponsive properties? Why were there two thermo-induced transitions observed? Which determines the thermoresponsive properties? Why does concentration have an effect on the behavior of macromolecules? Appropriate discussion should be added.

The English of the paper should be carefully checked.

I suggest citing relevant papers where similar systems were presented:

https://doi.org/10.1007/s00396-022-04959-1

https://doi.org/10.1021/acsami.2c20395

Author Response

Reviewer 2

Please suggest what types of buffers were used in the studies.

  1. The factor of the buffer solutions plays an important role in the pH-responsivity as well as the thermo-responsivety of the systems presented here. In a similar study [https://doi.org/10.1007/s00396-022-04959-1], it was shown that, in contrast to the "typical" behavior of POEGMA coatings in water, different mechanisms available in buffer solutions destroy hydrated layers surrounding POEGMA macromolecules, leading to their collapse.

The pH was varied from 3.6 to 12.4 in buffer solutions (pH 3.6, 6.86, 12.4, Hanna Instruments) and pH of water solution was determined by pH-meter (Sartorius, Finland) and pH-meter-ionomer Expert-001 (Russia).

Thank you very much for your comment. We have carefully reviewed the mentioned article. And we fully agree that in buffer systems (at lower pH values) systems collapse. This fact is confirmed that decreasing the phase separation temperatures T1 and T2 at low pH of buffer solutions.

  1. In contrast to the first section of the article, which does a great job of explaining the synthesis and structure of the polyOPG8OEG8MA-DMAPMA, questions about the molecule's behavior at various concentrations, temperatures, and pH levels keep coming up. I wholeheartedly concur that this system is extremely complex, but efforts should be made to explain the molecular mechanisms of temperature-induced phase separation (types of intramolecular and intermolecular bonds).

We explain the molecular mechanisms in the temperature range of phase separation by the intramolecular interaction, namely, the intramolecular bond formation and compactization, which is clearly seen in the Figure 5. That is, with a decrease in the size of scattering species, the excess intensity of scattered light decreases. And on further heating, along with compactization, intermolecular interactions occur, which lead to the formation of aggregates.

  1. Please provide tables 2, 3, and 4 with additional information. What do T1 and T2, ΔT and T ph, and other parameters mean?

T1 is the temperature of onset of phase separation in accordance with turbidimetry data.

T2 is the temperature of finishing of phase separation.

ΔT =T2-T1 is the width of phase separation interval.

Tph is the temperature of phase separation determined by transmitted spectroscopy.

All additional information are presented in section 3.3 «Characteristics of aqueous solutions of polyOPG8OEG8MA-DMAPMA on heating». But for the convenience of the readers, we rearranged the table below. So that the parameters are explained first, and then their values are presented in the Table 3, page 11.

  1. Unfortunately, in a lot of places, the authors state only phenomena without trying to describe their nature. Which determines the thermoresponsive properties? Why were there two thermo-induced transitions observed? Why does concentration have an effect on the behavior of macromolecules? Appropriate discussion should be added.

Thermosensitive properties are determined in the system by the main groups of the homopolymer and pH-sensitive monomer units of DMAPMA.  On heating, these systems lose their solubility.  With dilution, the quality of the solvent improves the size of formed aggregates at room temperature and at temperature T1 decreases. That is, the limits of solubility increase. This phenomenon is typical for solutions of thermo- and pH-sensitive polymers.

We do not observed  the change in macromolecule radii, since they are small. The aggregates decrease with dilution. These data are presented in the Table 4. page 13.

  1. The English of the paper should be carefully checked. I suggest citing relevant papers where similar systems were presented: https://doi.org/10.1007/s00396-022-04959-1 https://doi.org/10.1021/ acsami.2c20395.

English we have checked. Thank you. In Introduction we refer to articles https://doi.org/10.1007/s00396-022-04959-1 and https://doi.org/10.1021 /acsami.2c20395.  Pages 2 and 3.

All changes in the article are highlighted in yellow

Reviewer 3 Report

The manuscript by Simonava et al., describes thermo- and pH-responsive copolymer polymer brushes based on [PPG)-block-[PEG]-methacrylate with different percentages of N-[3-(dimethylamino) propyl] methacrylamide (from 0 to 20 %) synthesized via RAFT polymerization. Static, Dynamic light scattering techniques, viscometry and molecular modelling techniques are used to characterize the materials. Analyzing critical micelle concentrations that decrease with the presence of N-[3-(dimethylamino)propyl]methacrylamide indicates hydrophilic-lipophilic balance. The molecular brushes demonstrated thermo- and pH-responsiveness in water-salt solutions in conjunction with variation in contents of N-[3-(dimethylamino)propyl]methacrylamide units. The polymeric system is well described with regard to its chemistry and properties and this work can be accepted after addressing the following comments.

  1. The authors have used ‘’masse’’ instead of mass in some parts of the manuscript. Please recheck.

  2. What is the motivation behind polymer brush design? In the introduction, please explain the objective.

  3. Why DMFA and acetonitrile along with water were used specifically in this study? I understand that the ultimate application is in water. Does DMFA (and acetonitrile) have any solubility parameters that are suitable for this particular polymer system?

  4. I suggest including ‘’polymer brush’’ in the key words and avoiding ‘’optics’’ from there as I feel there is no direct experimentation related to optical application here.

  5. The polymer system is missing characterization data, such as 1HNMR and GPC. Please include the details either in the main paper or in a supplementary material.

  6. In line 215 on page no. 220, what does "CPC" mean.

  7. The majority of tables lack significant foot notes describing the elaboration of parameters and the method used. Eg. What is A0 in table 1. How did you find it?

  8. Table 1 does not include the parameters obtained in water as solvent. Please include those data to provide more information for readers.

  9. There are a lot of spell checks required throughout the manuscript. In FIGURE 2, it is mentioned ‘’schema’’. Please check the draft thoroughly.

  10. Is the thermoresponsiveness behaviour like LCST behaviour by NIPAM polymer? If so, please try to incorporate the term in the manuscript with suitable references so that it becomes clear for the readers.

  11. What about the reversibility of the trend in Figure 3 on cooling back to room temperature?

  12. The authors claim both molecular brush formation and micelle formation. It is suggested to show some microscopic data like AFM or TEM to prove the same.

Author Response

Reviewer 3

  1. The authors have used ‘’masse’’ instead of mass in some parts of the manuscript. Please recheck.

Thank you very much for your comment. We have corrected the text of the article.

  1. What is the motivation behind polymer brush design? In the introduction, please explain the objective.

The investigation is devoted to the development of new types of biocompatible polymeric stimuli-sensitive molecular brushes with adjustable conformational and phase transitions in aqueous solutions, the micelles of which can be used as nanocontainers for the delivery of hydrophobic drugs. It is offered to use new water-soluble macromonomers, such as esters of methacrylic with a diblock alcohol part containing hydrophilic oligoethylene glycol and hydrophobic oligopropylene glycol blocks for synthesis of polymeric brushes. A distinctive feature of such macromonomers is the possibility of fine regulation of their diphilic nature by varying the length and order of the arrangement of hydrophilic and hydrophobic blocks, which should be expressed in the manifestation of the amphiphilic properties of polymers on the basis of these macromonomers. N-dialkylaminoalkyl(meth)acrylamides whose polymers exhibit thermo- and pH-sensitive properties in aqueous solutions will also be used to form the main chain of polymeric brushes. The use of the proposed macromonomers will make it possible to obtain both molecular brushes with a usual core-shell structure and with a new, previously not described conformation, in which the shell of a molecular micelle will consist of “loops” of hydrophilic blocks of the macromonomer, which should be reflected in the behavior of molecular brushes in solutions.  In addition, the use of methacrylic macromonomers containing respectively one or two identical diblock oligoethylene glycol-oligopropylene glycol substituents will make it possible to obtain brushes with different densities of substituents in the macromolecules. This will expand the synthetic possibilities of forming polymer nanocontainers with desired properties. The study of the interaction of the obtained polymers with hydrophobic low molecular weight substances (models of drug compounds) in aqueous and water-salt solutions, including the capacity of polymer brushes and their aggregates with respect to the loaded model compound, as well as the conditions and kinetics of release of such compounds from a polymer nanocontainer, have practical significance and will have novelty through the use of new polymer containers. 

 We added the motivation of synthesis of copolymers in introduction, page 3.

  1. Why DMFA and acetonitrile along with water were used specifically in this study? I understand that the ultimate application is in water. Does DMFA (and acetonitrile) have any solubility parameters that are suitable for this particular polymer system?

In the study of homopolymer systems polyOPG6.6OEG8.3MA, the chloroform, THF, water, acetonitrile were used. It was shown that using of acetonitrile makes it possible to obtain molecularly dispersed solutions. In this solvent we determined molar mass and hydrodynamic radii. In chloroform and water the micelle formation was detected. This fact was confirmed by fluorescent method (CMC).  When comparing the sizes of micelles and macromolecules, it was found that micelles formed in chloroform consist of 2-3 macromolecules. Unfortunately, THF is not a very convenient solvent for light scattering studies because its characteristics change under light. So in present article which devoted to investigation copolymers polyOPG8OEG8MA-DMAPMA the same solvents were used to compare the results obtained for these systems.

The important task our study was to prove the additivity of copolymer blocks with different fraction of pH-sensitive component DMAPMA. That is, long-term experiments on refractometry are necessary. In DMFA, one mode was also fixed, but the value of the increment of the refractive index was low, however, in this solvent, we only confirmed the additivity of the increments of the refractive index of the components of the block copolymer.

  1. I suggest including ‘’polymer brush’’ in the key words and avoiding ‘’optics’’ from there as I feel there is no direct experimentation related to optical application here.

We have included this term in the keywords and excluded optics. (Page 1,Line 31). We only note    that we used optical methods (static and dynamic scattering of light under investigation of polymer solution).

  1. The polymer system is missing characterization data, such as 1HNMR and GPC. Please include the details either in the main paper or in a supplementary material.

We presented characterization data (GPC, NMR) in the main text of the article. Page 7

  1. In line 215 on page no. 220, what does "CPC" mean.

          CPC means GPC. This was a typo.  We have made changes to the article.

  1. The majority of tables lack significant foot notes describing the elaboration of parameters and the method used. Eg. What is A0 in table 1. How did you find it?

          The hydrodynamic invariant A0, which is calculated by the formula

This characteristic is determined by the experimental values of molar mass M, intrinsic viscosity [η] and translation diffusion coefficient D0 (η0 is viscosity of solvent).  The D0 values is determined using Stoker’s equation

D0 =kBT/f = kBT/6πη0Rh-D

 where k is the Boltzmann constant, and T is the absolute temperature.  Appropriate explanations have been added to the article. Page 10.

 This characteristic indicates a compact structure of the polymer and a decrease in the intramolecular density with an increase in the content of the DMAPMA component.

  1. Table 1 does not include the parameters obtained in water as solvent. Please include those data to provide more information for readers.

Table 1 presents the molar mass and hydrodynamic characteristics of polymer brushes determined in organic solvents, in which associative phenomena are not observed. In water, only micelles were observed and CMC was determined. Therefore we presented this information in Table 2, which shows data on the thermosensitivity of polymers in water solution. The size of scattering particles and MM determined in water are not corrected and do not described behavior of isolated macromolecules.

Nevertheless considering your wishes we included the values of hydrodynamic radii in Table 1, and we added corresponding information. Page 8

  1. There are a lot of spell checks required throughout the manuscript. In FIGURE 2, it is mentioned ‘’schema’’. Please check the draft thoroughly.

We carefully read the article and made all the necessary changes.

  1. Is the thermo responsiveness behavior like LCST behavior by NIPAM polymer? If so, please try to incorporate the term in the manuscript with suitable references so that it becomes clear for the readers.

Yes, the behavior of solutions of these polymers is similar to the behavior of polyisopropiacrylamide, only to determine the LCST is the task of the next study. So far, we have only shown that the phase separation temperatures decrease with increasing concentration, and in order to determine the LCST, a very wide concentration range, mainly large ones, is needed. Note, that in the region of high concentrations, the difference in phase separation temperatures is small.

  1. What about the reversibility of the trend in Figure 3 on cooling back to room temperature?

As our studies have shown, after cooling, the solutions had their original characteristics. The question of the reversibility of processes is very important from a practical point of view. The reversibility should strongly depend on the temperature and duration of the annealing. We are planning to conduct such studies in order to reveal the influence on the reversibility of the chemical structure of polymers.

  1. The authors claim both molecular brush formation and micelle formation. It is suggested to show some microscopic data like AFM or TEM to prove the same.

We have attempted to conduct AFM studies. Unfortunately, we were unable to obtain good samples for analysis due to their poor binding to the substrate. We continue research in this direction.

All changes in the article are highlighted in yellow

Round 2

Reviewer 2 Report

Now, the paper looks perfect and can be accepted for publication in its present form.